# Identification and Prevalence of Phascolarctid Gammaherpesvirus Types 1 and 2 in South Australian Koala Populations

**DOI:** 10.3390/v12090948

**Published:** 2020-08-27

**Authors:** Vasilli Kasimov, Tamsyn Stephenson, Natasha Speight, Anne-Lise Chaber, Wayne Boardman, Ruby Easther, Farhid Hemmatzadeh

**Affiliations:** School of Animal and Veterinary Sciences, The University of Adelaide, Roseworthy campus, 5371 Adelaide, Australia; vasilli.kasimov@hotmail.com (V.K.); tamsyn.stephenson@adelaide.edu.au (T.S.); natasha.speight@adelaide.edu.au (N.S.); anne-lise.chaber@adelaide.edu.au (A.-L.C.); wayne.boardman@adelaide.edu.au (W.B.); a1685587@student.adelaide.edu.au (R.E.)

**Keywords:** Phascolarctid gammaherpesvirus, koala, South Australia, PhaHV-1, PhaHV-2

## Abstract

To determine Phascolarctid gammaherpesviruses (PhaHV) infection in South Australian koala populations, 80 oropharyngeal swabs from wild-caught and 87 oropharyngeal spleen samples and swabs from euthanased koalas were tested using two specific PCR assays developed to detect PhaHV-1 and PhaHV-2. In wild-caught koalas, active shedding of PhaHV was determined by positive oropharyngeal samples in 72.5% (58/80) of animals, of which 44.8% (26/58) had PhaHV-1, 20.7% (12/58) PhaHV-2 and 34.5% (20/58) both viral subtypes. In the euthanased koalas, systemic infection was determined by positive PCR in spleen samples and found in 72.4% (63/87) of koalas. Active shedding was determined by positive oropharyngeal results and found in 54.0% (47/87) of koalas. Koalas infected and actively shedding PhaHV-1 alone, PhaHV-2 alone or shedding both viral subtypes were 48.9% (23/47), 14.9% (7/47) and 36.2% (17/47), respectively. Only 45.9% (40/87) were not actively shedding, of which 40.0% (16/40) of these had systemic infections. Both wild-caught and euthanased koalas actively shedding PhaHV-2 were significantly more likely to be actively shedding both viral subtypes. Active shedding of PhaHV-2 had a significant negative correlation with BCS in the euthanased cohort, and active shedding of PhaHV-1 had a significant positive relationship with age in both wild-caught and euthanased cohorts.

## 1. Introduction

The *herpesviridae* family contains widely prevalent double-stranded DNA viruses, classified into three subfamilies: *alphaherpesvirinae*, *betaherpesvirinae* and *gammaherpesvirinae*. They have been found to infect many species across the animal kingdom, including all mammalian and avian species investigated. The persistent and often lifelong infection of herpesviruses has allowed them to co-evolve with their animal hosts, which may have led to an adaptability advantage over other infectious diseases and contributed to the survival strategy of the virus [1].

Several herpesviruses have been characterised in Australian marsupials which include Macropodid herpesvirus-1 (MaHV-1, alphaherpesvirus) detected from oral and genital mucous membrane lesions in Parma wallabies (*Notamacropus parma*) during a mortality event in 1975 [2] Macropodid herpesvirus-2 (MaHV-2, *alphaherpesvirinae*) isolated from quokka kidney cells (*Setonix brachyurus*) [3], Macropodid herpesvirus-3 (MaHV-3, gammaherpesvirus) and Macropodid herpesvirus-4 (MaHV-4, *alphaherpesvirinae*) from a variety of tissues including whole blood, mammary covered gammaherpesviruses, Phascolarctid gammaherpesviruses-1 [4] (PhaHV-1) and PhaHV-2, detected in the liver, spleen and nasal scrapings from various koalas (*Phascolarctos cinereus*) [5,6,7]. Recently, Stalder et al. [8] conducted a surveillance study on a range of Australian marsupials (*n* = 278) and detected six additional novel herpesviruses (one alphaherpesvirus and five gammaherpesviruses); three in common wombats (*Vombatus ursinius*) (VoHV1–3), one in swamp wallabies (*Wallabia bicolor*) (MaHV-5), one in Tasmanian devils (*Sarcophilus harrisii*) (DaHV-2) and one in Southern brown bandicoots (*Isoodon obesulus*) (PeHV-1).

Gammaherpesviruses such as Epstein Barr virus (EBV) and Kaposi sarcoma-associated herpesvirus (KHV) in humans, ovine herpesvirus-2 (OVH-2) infections in sheep and cattle, and the recently discovered novel gammaherpesviruses found in koalas (PhaHV-1 and PhaHV-2) are lymphotropic by nature, initially infecting epithelial cells and then establishing a latent infection within B-cells and T-cells which are densely populated within the host spleen and lymph nodes [9]. These viruses typically lay dormant within host lymphocytes and prevent the cell from dying via the translation of effector proteins which interfere with natural cell pathways, enabling evasion of the host’s immune system. Eventually, due to external or environmental stressors, compromised immunity or other factors, a recrudescence of infection may occur causing the virus to suddenly replicate rapidly. This causes the cell to lyse, permitting new virions to be actively shed and be transmissible through respiratory and sexual transmission pathways [10,11,12].

Currently, PhaHV infection and prevalence has only been described in Victorian koala populations. PhaHV-1 and 2 were detected in 10.1% (10/99) and 23.2% (23/99) of surveyed koalas, respectively, with only one koala (1/99) being co-infected with both viral subtypes. Vaz et al. (2019) [7] also conducted a survey on 810 koalas from various populations across Victoria, with PhaHV prevalence ranging from 1 to 55%. PhaHV DNA has been detected from conjunctival, nasal, oropharyngeal, cloacal and prepuce swabs [8].

The clinical significance of PhaHV is still under scrutiny, with correlations reported between PhaHV infection and “wet bottom” in koalas, a clinical manifestation of *Chlamydia pecorum* infection [7,8]. No other direct correlations between disease and PhaHV infection have been reported in koalas. Vaz et al. [5] describe severe lymphoid depletion in both lymph nodes and spleen of infected koalas. These koalas also had other comorbidities, such as chronic dermatitis caused by *Sarcoptes scabiei*, chronic interstitial nephritis and cystitis, bilateral conjunctivitis, pulmonary congestion, enlarged nodular spleens and airway haemorrhages [5]. Splenic lymphoid area has been reported to be positively associated with koala retrovirus (KoRV) viral loads, and disease-free koalas have been shown to have small numbers or absence of periarteriolar lymphoid sheaths or splenic lymphoid follicles [13]. Cystitis and conjunctivitis are common findings in clinical chlamydiosis [14]. Conditions observed in PhaHV-positive koalas are similar to those observed during the Macropodid herpesvirus 1 (MaHV-1) outbreak in 1975 [2], in which infected wallabies displayed signs of conjunctivitis, pneumonia, splenic and hepatic necrosis [2,3].

Given the relatively high prevalence of PhaHV in Victoria, we hypothesised that both viral subtypes occur and are actively being shed within the South Australian koala population. Moreover, we expected that a significant relationship exists between age and PhaHV infection, due to the lifelong nature of infection and the increased likelihood of infection with time. We also expected there to be a relationship between poor body condition of koalas and shedding of the virus, since stress and being immunocompromised increases the likelihood of recrudescence in other species, resulting in active shedding, including from the oropharynx [15,16,17]. Most gammaherpesviruses shed virions from epithelial cells, therefore DNA detected from swabs of these tissues is most likely to confirm active shedding [18]. The latent stage of infections occurs within lymphocytes, which are also densely populated within the spleen [19,20], therefore infection status could be determined through DNA extracted from spleen samples.

The primary objectives of this study were to determine if either of the PhaHV viral subtypes were prevalent in South Australian koala populations, the percentage of koalas actively shedding the virus and whether active shedding of either viral subtype has significant correlations with factors such as age, body condition score or sex.

## 2. Materials and Methods 

### 2.1. Animal Ethics

This study was approved by the University of Adelaide Animal Ethics Committee and conducted in accordance with the guideline set out in the Australian Code for the Care and Use of Animals for Scientific Purposes 8th Edition (2013) (National Health and Medical Research Council: Canberra, 2013). Animal ethics approval number for wild-caught koalas: S-2018-022 (granted 6 April 2018) and for the euthanased examination cohort: S-2016-169 (granted 9 January 2017); DEW scientific permit number: Y26054-7 (granted 7 September 2017).

### 2.2. Sample Collection

Two cohorts of koalas were used for this study: wild-caught and euthanased koalas, both from the Mount Lofty Ranges koala population in South Australia. The wild-caught cohort was considered to represent a random sample to investigate active shedding of virus, and the euthanased cohort, euthanased on welfare grounds, enabled investigations of systemic PhaHV infection status.

### 2.3. Wild-Caught Koalas

As part of a larger koala health surveillance project, wild-caught koalas were sourced from three national parks or reserves in the Mount Lofty Ranges, in Morialta, Cleland and Belair. They were caught by the flag technique, which utilized a large pole with a flag attached to the far end. The flag end of the pole was used to direct the koala down the tree. Once the koala was close to the ground, it was restrained and taken to a field hospital where it was anaesthetised and samples collected, including oropharyngeal swabs. Oropharyngeal swabs were used for PhaHV testing due to their increased rate of positivity [8] and suitability in the field. Further demographic data were recorded and included tag identification, sex, tooth wear class (TWC I-VII) [21] and body condition score (BCS 1–5) [22]. Global positioning system (GPS) data and tagged trees were recorded so koalas could be released at their point of capture. Oropharyngeal swabs were placed in sealed plastic bags and kept on ice before storage at −80 °C within 12 h of sampling. Samples were kept in the −80 °C freezer for up to 12 months until tested.

### 2.4. Euthanased Cohort

For koalas that had been euthanased on welfare grounds, sex, TWC [21] and BCS [22] were recorded. Oropharyngeal swabs and spleen samples were collected and stored at −20 °C until tested. Spleen samples were collected to determine systemic infection status due to gammaherpesvirus latency in immunological cells [11,12,18,23,24].

### 2.5. DNA Extraction

DNA was extracted from both oropharyngeal swabs and spleen tissue samples using the QIAamp DNA Mini Kit (QIAGEN, Hilden, Germany). The concentration of the extracted DNA was measured using the NanoDrop One Spectrophotometer (Thermo Fisher Scientific Inc, Waltham, MA, USA). A working solution of 20 ng/µL of DNA from the extracted stock solutions was prepared for PCR tests.

### 2.6. Quality Control

The koala beta (β)-actin gene was screened via qPCR, as a quality control, from extracted oropharyngeal and spleen DNA samples to confirm adequate DNA was extracted, adopting the same protocol described by Shojima et al. [25]. Any samples negative for β-actin were removed from the study due to a lack of quality for further testing. DNA samples were run in triplicate in a 5 µL reaction. The DNA copy number was derived from a standard curve from the purified PCR product from a South Australian koala. Negative control contained no DNA template.

### 2.7. Molecular Diagnostics (Conventional PCR)

Specific primers were designed for PhaHV-1 and PhaHV-2 based on the published DPOL gene (Table 1), due to both viral subtypes only having a 60% nucleotide pairwise identity [6]. Primers were confirmed to be specific via NCBI Primer Blast, Sanger sequencing and by testing the primer sets on each of the PhaHV subtypes (Table 2). PCR reactions were run in 20 µL of volume which included 0.5 µM of forward and reverse primer, 5 µL of 4× AllTaq Master Mix solution (QIAGEN, Hilden, Germany), 5 µL of 20 ng/µL DNA template and 8.5 µL of ultrapure water. PCR conditions were initial activation and denaturation of 95 °C for 2 min, followed by 34 cycles of denaturation at 95 °C for 5 s, annealing at 61 °C (PhaHV-1) or 64 °C (PhaHV-2) for 15 s and extension at 72 °C for 10 s. This was followed by a final extension step of 72 °C for 10 s.

### 2.8. Statistical Analyses

Binary Logistic Regression analyses (performed using IBM SPSS Statistics 23) were used to determine any significant relationships between infection of either PhaHV subtype, coinfections of PhaHV, BCS, TWC and sex. Compromised BCS was considered as BCS 1 to 3 out of 5 (1–3/5) (emaciated, poor, or fair muscle condition), and good BCS as 4–5/5. Variables with *p*-values of ≤0.05 were considered statistically significant.

A Pearson correlation coefficient (r) analysis was conducted on both wild-caught and euthanased cohorts (*n* = 80 and *n* = 87, respectively) to show correlations between covariates. Covariates analysed within each cohort were as follows; within the euthanased cohort: infected (PhaHV DNA detected in the spleen); Oro_Phas1, Oro_Phas2 and Oro_coinfection (PhaHV-1 DNA only, PhaHV-2 DNA only, or both PhaHV subtypes DNA detected from oropharyngeal swab, respectively); Spleen_Phas1, Spleen_Phas2 and Spleen_coinfection (PhaHV-1 DNA only, PhaHV-2 DNA only, or both PhaHV subtypes DNA detected from spleen tissue, respectively); TWC (tooth wear class) [21]; BCS (body condition score) [22], BCS_compromised (BCS 1–3/5) and sex. Covariates within the wild-caught koalas: infected (PhaHV-1 or PhaHV-2 DNA detected from oropharyngeal swab), Oro_Phas1, Oro_Phas2 and Oro_coinfection, TWC; BCS; BCS_compromised and sex. The results were visualized on a correlation heatmap using R version 3.0.1 using ggplot2 and ggcorrplot packages.

## 3. Results

### 3.1. PhaHV-1 and PhaHV-2 Specific PCR Test

The newly designed primer sets designated, VK-PhaHV-1 and VK-PhaHV-2, were both specific in detecting PhaHV-1 and PhaHV-2, respectively. Samples were confirmed in Sanger Sequencing (Australian Genome Research Facilities), both PhaHV-1 and PhaHV-2 showing 100% identity to GenBank sequence identifications (Table 2) (accession numbers: JN585829.1, JQ996387.1).

### 3.2. Wild-Caught Cohort of Koalas

Approximately three-quarters, 72.5% (58/80), of the wild-caught koalas from the Mount Lofty Ranges were actively shedding PhaHV. These were shown to be actively shedding just PhaHV-1 (44.8% (26/58)), just PhaHV-2 (20.7% (12/58)) or both viral subtypes (34.5% (20/58)). Only 27.5% (22/80) of the wild sampled koalas were not actively shedding either viral subtype (Table 3).

Infection of PhaHV-1 only had a significant positive correlation with TWC (*p*-value = 0.023; r = 0.2; *n* = 80) (Figure 1A). The proportion of koalas in each TWC for each status of infection is shown in Figure 2. There were no significant associations between splenic infection with PhaHV and koalas with a compromised BCS (*p*-value = 0.203; *n* = 80) or with sex (*p*-value = 0.776; *n* = 80).

### 3.3. Euthanased Cohort of Koalas

It was found that 72.4% (63/87) of euthanased koalas were infected with PhaHV. Active shedding of the virus occurred in 54.0% (47/87) of koalas in the euthanased cohort, with active shedding of only PhaHV-1 in 48.9% (23/47), only PhaHV-2 in 14.9% (7/47), or both subtypes simultaneously in 36.2% (17/47) of koalas. All koalas had matched positive spleen samples which confirmed infection (Table 3). Furthermore, 18.4% (16/87) of koalas were systemically infected with the virus but not actively shedding virus. The remaining 27.6% (24/87) of koalas were not infected at either the oropharyngeal or splenic site.

There was a strong correlation between splenic coinfection with both PhaHV-1 and PhaHV-2, as being co-infected significantly increased the likelihood of actively shedding (positive oropharyngeal sample) both viral subtypes (*p*-value < 0.01; r = 0.80; *n* = 87) (Figure 1B). A significant positive correlation exists between active PhaHV-1 and PhaHV-2 infection, as koalas were 3.5 times more likely to be actively shedding PhaHV-1 if infected with PhaHV-2 (*p*-value = 0.006; r = 0.3; Exp (B) = 3.538) (Figure 1B). There was a high probability of koalas actively shedding PhaHV-1 (Exp (B) = 28.51) and PhaHV-2 (Exp (B) = 92.49) when co-infected with both viral subtypes in the spleen (Figure 3). Similarly, to the wild-caught koalas, the euthanased cohort had a significant positive correlation between TWC and PhaHV-1 infection (Figure 1A,B) (*p*-value = 0.05; r = 0.2; *n* = 87). However, contrary to wild-caught koalas, a significant positive correlation exists between active shedding of PhaHV-2 and a compromised BCS (*p*-value = 0.04; r = 0.2; *n* = 97) (Figure 1B), with euthanased koalas actively shedding PhaHV-2 being 3.5 times more likely to have a compromised BCS (Exp (B) = 3.514). No significant association between compromised BCS and active shedding of PhaHV-1 or sex was found (*p*-value = 0.2 and 0.16, respectively).

## 4. Discussion

In this study, oropharyngeal and spleen samples were tested, as previous papers [5,6] succeeded in detecting and isolating the same viral subtypes from these sites. In the wild-caught cohort, only oropharyngeal swabs could be collected, whereas in the euthanased cohort, both oropharyngeal and spleen could be collected. As previously mentioned, it has been assumed that the PCR-positive oropharyngeal swabs are most likely to confirm active shedding [18], whilst latent infections occur within lymphocytes which are densely populated within the spleen [19,20]. In the euthanased cohort, koalas that were positive for PhaHV in both spleen tissue and oropharyngeal swab samples were deemed as infected and actively shedding the virus. Koalas only positive in the spleen were classified as being infected.

### 4.1. Active Shedding

In this study, the prevalence of PhaHV active shedding in wild-caught koalas was 72.5% (58/80) and was higher than that of the euthanased koalas at 54% (47/87). The rate of active shedding was higher than initially expected when compared with other herpesvirus cases found in other koalas in Australia. Stalder et al. [8] surveyed 99 captive and wild koalas in Victoria in 2015 and detected PhaHV-1 and 2 in 10.1% (10/99) and 23.2% (23/99) of koalas, respectively. Another study in Victoria found the prevalence of PhaHV-1 and PhaHV-2 was 17% and 22%, respectively, and an overall prevalence of 33% (*n* = 810), with different populations ranging from 1 to 55% [7]. Mainland populations in Victoria had 22–46% and 31–55% test positive for PhaHV-1 and PhaHV-2 [7], respectively. These numbers are more consistent with the prevalence in Mount Lofty Ranges koalas, with active shedding of PhaHV-1 and PhaHV-2 at 57.5% (46/80) and 40% (32/80), respectively, in the wild-caught koalas and 46% (40/87) and 27.5% (24/87), respectively, in euthanased koalas. These may represent populations with similar high densities with greater opportunity for infection spread. Differences in prevalence of viral shedding between PhaHV-1 and PhaHV-2 may show that PhaHV-2 is less established within the SA koala population than PhaHV-1, or that PhaHV-2 may have a lower tendency to actively shed than PhaHV-1. This may be due to the genetic differences between the viral subtypes, as PhaHV-2 only shares 60% pairwise identity with PhaHV-1 [6] and may contribute to higher expression of immune suppressor genes in PhaHV-2. Latency studies would need to be carried out for this theory to be concluded.

A significant relationship and positive correlation existed between active shedding of PhaHV-1 and TWC in both the wild-caught cohort (*p*-value = 0.023, r = 0.3) and the euthanased cohort (*p*-value = 0.05, r = 0.2), whereby increasing age increased the likelihood of PhaHV-1 infection. This relationship was expected: as age increased, the probability of infection increased due to the likelihood of exposure to the virus and the lifelong nature of the disease. In contrast, there was no correlation between active shedding of PhaHV-2 and TWC in either cohort. The study by Vaz et al. [7] also showed similar results regarding PhaHV-1 infection correlation with age and PhaHV-2 spread more evenly across age groups. This study proposed that PhaHV-1 infection may be more likely due to sexual contact between adult koalas, whilst PhaHV-2 infection may occur from close contact between mother and joey [7]. Vaz et al. showed repeatable detection of both PhaHV-1 and PhaHV-2 in koala cloacal regions [7,8], therefore, viral transmission between mother and joey may not only occur in the pouch, but potentially during parturition and/or from the consumption of a unique maternal faeces known as “pap”. This faecal complex contains tannin protein complex-degrading enterobacteria (T-PCDE) essential for digestion of gum leaves [26] and may also be contaminated with other diseases such as *Chlamydia pecorum* and PhaHV-1 and PhaHV-2. Kent et al. [27] conducted a survey on sexually transmitted mustelid gammaherpesvirus 1 (MusHV-1) in European badgers (*Meles meles*) and investigated the prevalence of infection in adults and cubs; the high proportion of infected cubs showed a strong likelihood of vertical transmission.

Body condition score is a commonly used and effective indicator of the health status of an animal [18]. In our study, koalas with a BCS of 1–3/5 (emaciated, poor, or fair muscle condition) were considered compromised, whilst koalas with BCS 4–5/5 were considered in good condition. It was hypothesised that koalas with a low BCS were more likely to be actively shedding if infected, due to being unable to keep the herpesvirus in a latent, suppressed state. Active shedding of PhaHV-2 had a significant correlation with a compromised BCS in the euthanased koala cohort (*p*-value = 0.04; r = 0.2; *n* = 87), but not in the wild-caught cohort. This was likely confounded by the spread of koalas in each BCS category with a right-skewed bias towards healthy koalas in the wild-caught cohort and a more normal distribution in the euthanased cohort (Figure 4). Surprisingly, compromised BCS had no significant correlation with infection or active shedding of PhaHV-1 in either cohort. The lack of relationship between compromised BCS and active shedding of PhaHV-1 is likely due to apparently healthy animals being able to actively shed herpesviruses without developing clinical disease [10]. The potential to asymptomatically shed gammaherpesviruses or have subclinical disease has been demonstrated in other reservoir hosts [11]. In a case study of ovine herpesvirus-2 (OvHV-2; gammaherpesvirus), Li et al. [10] took a series of nasal swabs and discovered that healthy lambs between 6 and 9 months of age had an active shedding prevalence of 61% (*n* = 56). Gammaherpesviruses can be more likely to cause disease in susceptible, nonreservoir hosts, as seen with cattle infected with OvHV-2 causing malignant catarrhal fever (MCF) [28].

### 4.2. Systemic Infection and Active Shedding

The majority of infected euthanased koalas were actively shedding the virus (74.6%; 47/63), whilst only 25.4% (16/63) of infected koalas were not actively shedding. There was a strong positive correlation between splenic coinfection of both viral subtypes and active shedding of the virus (0.8) (Figure 1B). In the euthanased cohort, koalas infected with PhaHV-2 were 3.5 times more likely to be coinfected with PhaHV-1 (Exp (B) = 3.538; *p*-value = 0.006; *n* = 87) and as a result were 1.5 times more likely to be actively shedding PhaHV-1 with every incremental increase in TWC (Exp (B)) = 1.514; *p*-value = 0.05; *n* = 87). This suggests that viral coinfection highly predisposes koalas to active shedding of the virus. Potentially, increased expression of viral effector proteins from both PhaHV-1 and PhaHV-2 could lead to greater immunosuppression, initiating recrudescence. A study investigating herpesvirus coinfection in humans showed that infection of EBV and HHV-7 were also shown to promote HHV-6 infection and disease [29,30,31,32] however, this may just be a result of severe immunosuppression that triggers viral coinfection, as stated by Handous et al. [33]. PhaHV-2 had a significant correlation with a compromised BCS, suggesting that PhaHV-2 may be an indicator of other underlying health conditions such as immunosuppression, stress and coinfection of other diseases including PhaHV-1, *Chlamydia pecorum* or koala retrovirus.

In our study, there was no significant relationship between sex and infection status. Yet Stalder et al. [8] showed a significant relationship between infection and sex, with male koalas more likely to have an infection. Vaz et al. [7] showed that females without young were more likely to be infected than those with young; they also found a correlation between PhaHV and “wet bottom”, a clinical sign of overt chlamydiosis. The association of potentially infertile females with PhaHV may be confounded by chlamydial infection, a known cause of reproductive pathology in koalas [34].

### 4.3. Sites of Active Viral Shedding

In our study, most oropharyngeal swabs from infected (spleen positive) koalas were positive for Phascolarctid gammaherpesvirus, suggesting that significant active shedding is from rostral epithelial tissues (nasal, oropharyngeal). This finding differs to that of Vaz et al. [7], which found that rostral (ocular, nasal, oropharyngeal) swabbing was less likely to pick up an infection in comparison with caudal (urogenital, cloacal) swabbing. Samples from caudal epithelial tissues were not tested in our study, therefore, shedding from different epithelial sites needs to be investigated further to provide tissue tropism insights. One concern about the interpretation of active shedding is the contamination of the oropharyngeal swabs with cells within which latency has been established, that is, circulating mononucleocytes. Nevertheless, due to the tissue tropism of herpesviruses, it is less likely to detect virus in epithelial swabs when the animals are only infected in latent forms [15,35,36]. It is also highly likely that the detected viruses in epithelial cells are actively shed viruses [37].

### 4.4. Potential Areas for Further Research

This study focused on the presence and prevalence of Phascolarctid herpesviruses within the mainland South Australian koala population. Further research is needed to understand the association of PhaHV with other infections—primarily koala retrovirus (KoRV) and Chlamydial infection—since PhaHV may be playing a role in the augmentation of clinical disease or be shed secondary to them. Since PhasHV-2 had an association to koalas with a compromised BCS and increased PhasHV-1 shedding, this could be suggestive of an increase in pathogenicity in this subtype. Investigation into disease presentations, haematology and immune function markers with and without PhaHV infection may shed some light on the effect of these viruses. Serological studies could be conducted to investigate the koalas’ response to PhaHV infection, since the koalas’ immune system has often been perceived as “immunologically lazy” [38,39]. These studies would help to further determine the clinical significance, host response to infection and impact of these newly discovered viruses on koala populations.

## 5. Conclusions

We showed there was a high prevalence of PhaHV infection in koalas in the Mount Lofty Ranges population, with more than two-thirds of both wild-caught and euthanased cohorts actively shedding the virus. PhaHV-1 had a greater prevalence within the SA koala populations, with more koalas actively shedding PhaHV-1 than PhaHV-2.

Despite being less prevalent, koalas coinfected with PhaHV-2 were more likely to also be infected and actively shedding PhaHV-1. Neither of the viral subtypes were shown to have any significant relationship with BCS in wild-caught koalas; however, PhaHV-2 infection had a significant correlation with BCS in euthanased koalas. PhaHV-1 infection was also shown to be positively correlated with TWC in both cohorts, whilst sex had no significant correlation with either viral subtype in both cohorts.

The clinical significance of these recently discovered Phascolarctid herpesviruses is still unknown, and additional investigation into the pathogenicity, clinical signs of the virus and coinfection with other pathogens is important. Uncovering the significance of PhaHV will help determine the health status and guide the management of koala populations.

## Figures and Tables

**Figure 1 viruses-12-00948-f001:**
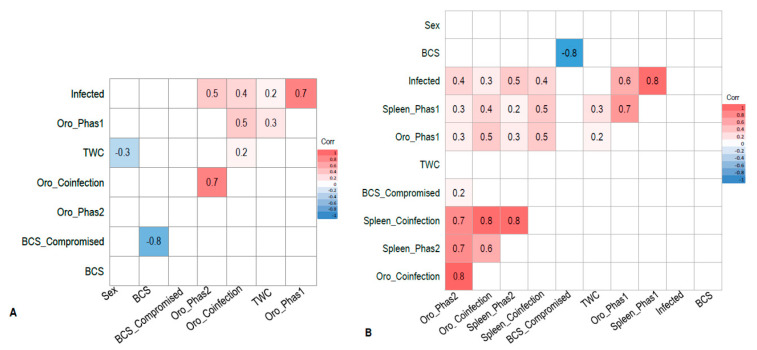
(**A**) Pearson’s correlation matrix displaying interactions between covariates in wild-caught koalas from the Mount Lofty Ranges, South Australia (*n* = 80) portrayed on a heat map. Values closer to 1.0 indicate a stronger positive correlation between the two variables. Coloured tiles contain a *p*-value ≤ 0.05. White tiles correspond to an insignificant correlation (*p*-value > 0.05) and are excluded from the model. Variable “BCS_compromised” uses compromised koalas (BCS 1–3/5) as the reference value. (**B**) Pearson’s correlation matrix displaying interactions between covariates from euthanased koalas sourced from various wildlife hospitals in Adelaide (*n* = 87), portrayed on a heat map. Values closer to 1.0 indicate a stronger positive correlation between the two variables. Coloured tiles contain a *p*-value ≤0.05. White tiles correspond to an insignificant correlation (*p*-value > 0.05) and are excluded from the model. Variable “BCS_compromised” uses compromised koalas (BCS 1–3/5) as the reference value. Acronym definitions are provided in Section 2.8 (statistical analysis).

**Figure 2 viruses-12-00948-f002:**
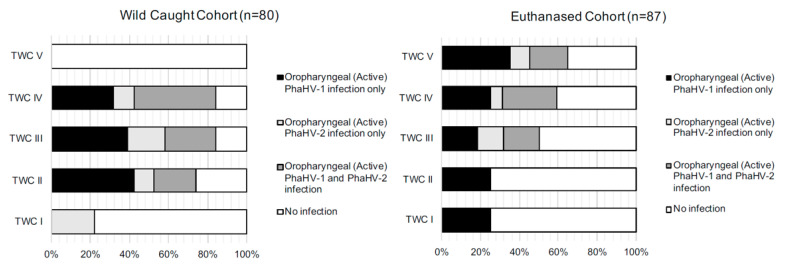
Histograms displaying the percentage of koalas actively shedding PhaHV-1 and PhaHV-2 and noninfected koalas within each tooth wear classification (TWC) group, for wild-caught and euthanased cohorts from South Australia.

**Figure 3 viruses-12-00948-f003:**
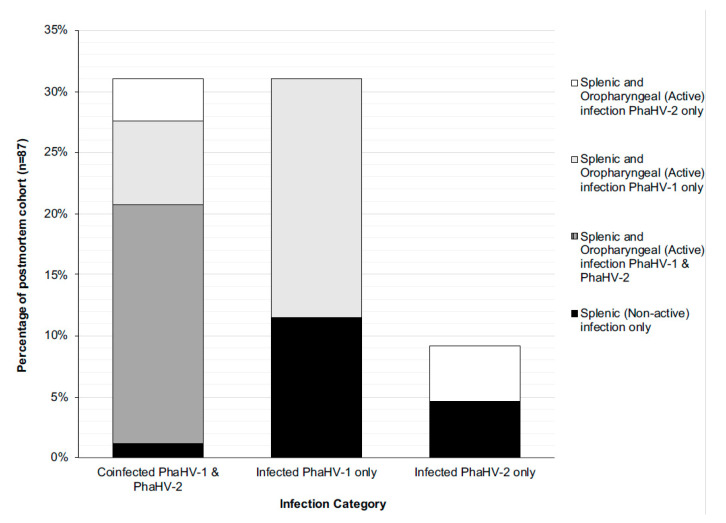
Histogram of infection status within the euthanased cohort (*n* = 87), displaying the following categories: coinfection with both viral subtypes (demonstrated by a dual splenic infection, and actively shedding both or singular viral subtypes, or not shedding); infected with just PhaHV-1 (demonstrated by a PhaHV-1 splenic infection and shedding or not shedding PhaHV-1); infected with just PhaHV-2 (demonstrated by a PhaHV-2 splenic infection and shedding or not shedding PhaHV-2).

**Figure 4 viruses-12-00948-f004:**
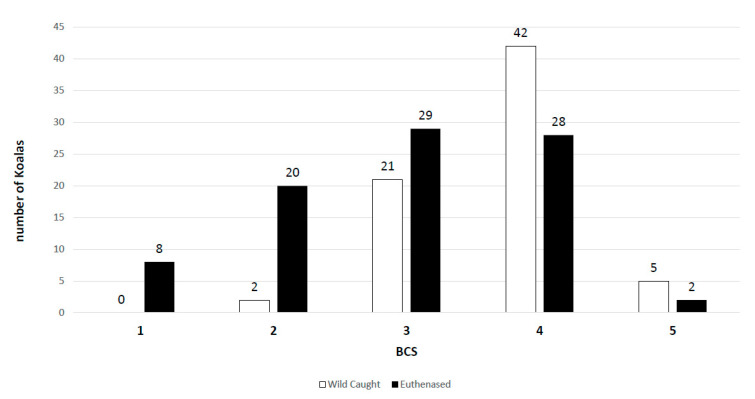
Histogram displaying the distribution of koalas in each cohort within their assigned body condition score (BCS).

**Table 1 viruses-12-00948-t001:** PCR primers, products and annealing temperatures.

Target	Primer Name	Primer Sequence	Product Bp	Annealing Temp	Reference
β-actin (QC)	β-actin-Fwd	5′ GAGACCTTCAACACCCCAGC 3′	111	60 °C	Shojima et al. (2013) [25]
	β-actin-Rev	5′ GTGGGTCACACCATCACCAG 3′			
PhaHV-1	VK-PhaHV-1-Fwd	5′ CGGCATCCTCCCCTGTTTAA 3′	220	61 °C	Current study
	VK-PhaHV-1-Rev	5′ GCCCCTACATTCAACGAACA 3′			
PhaHV-2	VK-PhaHV-2 Fwd	5′ CGCACTCTAAGCTGTCCCTT 3′	330	64 °C	Current study
	VK-PhaHV-2 Rev	5′ TTTCGAGCATCATGCGTCCT 3′			

**Table 2 viruses-12-00948-t002:** Results from AGRF Sanger sequencing, showing primer sets used, sample number, type of sample (oropharyngeal or spleen), query and identity to published GenBank sequences (accession numbers: JN585829.1, JQ996387.1).

Primers	Sample	PhaHV-1 (JN585829.1)	PhaHV-2 (JQ996387.1)	Source	Query Cover	Per Ident
VK-PhaHV-1	K18-051	+		Oro	100%	100%
	K18-051	+		Spleen	100%	100%
	K18-064	+		Oro	100%	100%
	K18-064	+		Spleen	100%	100%
VK-PhaHV-2	K18-043		+	Oro	100%	100%
	K18-043		+	Spleen	100%	100%
	K18-044		+	Oro	100%	100%
	K18-044		+	Spleen	100%	100%

**Table 3 viruses-12-00948-t003:** Prevalence of active shedding of Phascolarctid gammaherpesvirus (PhaHV) viral subtypes in the two cohorts, wild-caught and euthanased.

Type of Infection	Wild-Caught	Euthanased
*n*	%	*n*	%
Active Shedding	58	73%	47	54%
Active Shedding Only PhaHV-1	26	33%	23	26%
Active Shedding Only PhaHV-2	12	15%	7	8%
Coinfected Shedding	20	25%	17	20%
No active shedding	22	28%	40	46%
TOTAL	80		87

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
