# Peer review of "Identification and Prevalence of Phascolarctid Gammaherpesvirus Types 1 and 2 in South Australian Koala Populations"

_viruses, 2020, doi:10.3390/v12090948_

Round 1

Reviewer 1 Report

The paper is well written and organized and presents new information about gammaherpesvirus infection of an important, charismatic species in Australia.

It would be helpful to the reader to underscore the importance of doing the study by indicating why it is necessary for  understanding the influence that these two gammaherpesviruses may have on the health of koala populations and their management. A necessary first step in that understanding is determining how frequently these infections occur among koalas, the proportion that shed virus and prevalence of infection by age and BCS.

The authors point out reasonably that PhaHV-2 had a significant correlation with compromised BCS in the and may have been an indictor of poorer health condition in the euthanized koalas. They do not mention if PhaHV-2 infection itself could have contributed to the reduced BCS. A comment would be helpful even though the clinical consequences of infection by these viruses is not known currently.

Detection of viral DNA in samples may or may not indicate the presence of infectious virus. Does the presence of virus DNA always indicate active virus shedding, given the life-long infection of lymphoid tissue? Does the amount of infectious virus shed vary over time? That would have a lot to do with the risk of horizontal virus transmission. A comment would be helpful.

Author Response

Response to reviewers for manuscript ID: viruses-889130

Title: Identification and prevalence of Phascolarctid gammaherpesvirus types 1 and 2 in South Australian koala populations.

Please find our responses in Italic to the reviewers’ comments:

Reviewer 1:

The paper is well written and organized and presents new information about gammaherpesvirus infection of an important, charismatic species in Australia.

It would be helpful to the reader to underscore the importance of doing the study by indicating why it is necessary for understanding the influence that these two gammaherpesviruses may have on the health of koala populations and their management.

A necessary first step in that understanding is determining how frequently these infections occur among koalas, the proportion that shed virus and prevalence of infection by age and BCS.

  • The available information on Phascolarctid gammaherpesvirus is very limited and the available publications for this group of viruses are mostly focused on the discovery and novelty of these viruses in Koalas or similar marsupial species.
  • Same as the majority of other gammaherpesvirus, we expected that the Phascolarctid gammaherpesvirus produce more latent infections and might not be connected to an obvious clinical symptoms (Hwang et all (2008))
  • We have tried to provide more information on clinical relevance of Phascolarctid gammaherpesvirus at the Introduction part of the manuscript (lines 34-64 of revised manuscript).

The authors point out reasonably that PhaHV-2 had a significant correlation with compromised BCS in the and may have been an indictor of poorer health condition in the euthanized koalas. They do not mention if PhaHV-2 infection itself could have contributed to the reduced BCS. A comment would be helpful even though the clinical consequences of infection by these viruses is not known currently.

  • Thanks for the constructive comments by the reviewer 1 on the relationship of BCS and the PCR results. We have tried to clarify our observations and recorded BCS with the Phascolarctid gammaherpesvirus condition. As it has explained it at the above response, it is hard to provide strong evidences on clinical relevance of the Phascolarctid gammaherpesvirus infection including BCS. It was our observation and we have recorded as possible connection between the infection and low BCS. We tried to make it clear in the Statistical Analysis and the results by explaining more on the observed BCS and the PCR results.

Detection of viral DNA in samples may or may not indicate the presence of infectious virus. Does the presence of virus DNA always indicate active virus shedding, given the life-long infection of lymphoid tissue? Does the amount of infectious virus shed vary over time? That would have a lot to do with the risk of horizontal virus transmission. A comment would be helpful.

  • That is again a valuable comment and it is always an existing argument between the virologists and epidemiologist on the sensitivity, specificity and predictive values of the diagnostic tests and the infection/ disease conditions.
  • In this particular case if, we compare the Phascolarctid gammaherpesvirus with the other gammaherpesvirus we can make this statement that we are dealing with a lymphotropic herpesvirus. In this case, the Phascolarctid gammaherpesviruses make both active and latent phases of the infection at the lymphatic tissues but shed only via secretions or excretions. That is why we have targeted the lymphatic tissues as the main source of the infection but the swabs as a source of shedding.
  • We made more clarifications to differentiate the infection and the shedding using the molecular diagnostic tools in our study. We believe that at the current situation, the molecular tools are the only available tools for detection of the Phascolarctid gammaherpesvirus, while we have no pathognomonic signs in this cases.

Reviewer 2 Report

This is an interesting article that provides disease surveillance data for wild-caught and captive koalas in South Australia. The main issues I have with the paper are the introduction, which lacks pertinent information about this virus and associated disease, and the (over)interpretation of the results, which needs some toning down as well as a complete justification of this interpretation of the data before the results are presented. Without detailed knowledge of the pathogenesis of these viruses, the authors are drawing several large conclusions based on PCR data alone. I would like to see all instances of such "reaching" or speculation removed/toned down in the manuscript. The methodology also need significant beefing up, especially if this is the first presentation of these PCR assays- if that is the case, the authors should reference and adhere to the MIQE Guidelines (Bustin et al. 2009) about the extent of data that should be presented for publication of a new PCR assay. Once those modifications are achieved, I think this will make a nice paper and an important addition to the literature. I have made detailed comments in the attached, marked-up pdf. 

Author Response

Response to reviewers for manuscript ID: viruses-889130

Title: Identification and prevalence of Phascolarctid gammaherpesvirus types 1 and 2 in South Australian koala populations.

Please find our responses in Italic to the reviewers’ comments:

Reviewer 2:

This is an interesting article that provides disease surveillance data for wild-caught and captive koalas in South Australia.

  • Thanks for the constructive comments and valuable suggestions for the revision of the manuscript. We believe by revising the papers based on the reviewer 2 suggestions, our paper has improved significantly.

The main issues I have with the paper are the introduction, which lacks pertinent information about this virus and associated disease, and the (over)interpretation of the results, which needs some toning down as well as a complete justification of this interpretation of the data before the results are presented.

Without detailed knowledge of the pathogenesis of these viruses, the authors are drawing several large conclusions based on PCR data alone. I would like to see all instances of such "reaching" or speculation removed/toned down in the manuscript.

  • We have tried in best possible way to revise the manuscript especially at the introduction part to provide more supporting references, more description on the relevance of the infection with the test results and finally we have modified some parts of the manuscript to “toning down” our message. Please see the revisions at the lines of 65-68, 89-92 and more descriptions at the methods and data analysis.
  • It has also discussed at the response to the reviewer 1 comments on the reliability of the tests and the nature of gammaherpesvirus. In this particular case we have make some modification to highlight the importance of the Phascolarctid gammaherpesvirus and it possible effects on the hosts and shedding the virus.

The methodology also need significant beefing up, especially if this is the first presentation of these PCR assays- if that is the case, the authors should reference and adhere to the MIQE Guidelines (Bustin et al. 2009) about the extent of data that should be presented for publication of a new PCR assay.

  • In this study, we have developed test, validate the results in multiple different ways, and followed the commonly accepted methods for test validation including the suggestion by the Bustin et al. 2009 publications. As it has discussed at the paper all of the PCR positive samples were double-checked in Sanger sequencing. At the beginning of the work, some known positive samples were used to modified and readjust the PCR condition. The limit of the detection and the sequencing in all of the tests were applied to make sure we employ the sensitive and specific tool in our study.
  • The Housekeeping gene PCR was also used in all samples to make sure the negative results are not due to low DNA quality of general PCR inhibitors. As it has explained the negative samples in HK PCRs were removed from the analysis.

Once those modifications are achieved, I think this will make a nice paper and an important addition to the literature. I have made detailed comments in the attached, marked-up pdf. 

  • All of the corrections have been revised accordingly based on the valuable reviewer’s comments and suggestions.

Round 2

Reviewer 2 Report

The manuscript is significantly improved. I have only minor suggested edits to help improve readability- see attached marked up draft 2. The addition of background info in the introduction addresses my previous comments about the interpretation of the data re: "active" vs. "systemic" infection. I think it is in pretty good shape! 

Author Response

Dear Editor

The manuscript has changed accordingly based on the reviewer's minor comments and corrections. The final revision and the version with the track changes have uploaded to the online submission system.

Regards

Farhid Hemmatzadeh